# Drone Detection Method Based on MobileViT and CA-PANet

**Qianqing Cheng** [1,*] , **Xiuhe Li** [1,*], **Bin Zhu** [1], **Yingchun Shi** [1] **and Bo Xie** [1]

School of Electronic Countermeasures, National University of Defense Technology, Hefei 230037, China
* Correspondence: chengqianqing@nudt.edu.cn (Q.C.); xhli75@163.com (X.L.); Tel.: +86-1552-803-7356 (Q.C.)

**Abstract:** Aiming at the problems of the large amount of model parameters and false and missing detections of multi-scale drone targets, we present a novel drone detection method, YOLOv4-MCA, based on the lightweight MobileViT and Coordinate Attention. The proposed approach is improved according to the framework of YOLOv4. Firstly, we use an improved lightweight MobileViT as the feature extraction backbone network, which can fully extract the local and global feature representations of the object and reduce the model's complexity. Secondly, we adopt Coordinate Attention to improve PANet and to obtain a multi-scale attention called CA-PANet, which can obtain more positional information and promote the fusion of information with low- and high-dimensional features. Thirdly, we utilize the improved K-means++ method to optimize the object anchor box and improve the detection efficiency. At last, we construct a drone dataset and conduct a performance experiment based on the Mosaic data augmentation method. The experimental results show that the mAP of the proposed approach reaches 92.81%, the FPS reaches 40 f/s, and the number of parameters is only 13.47 M, which is better than mainstream algorithms and achieves a high detection accuracy for multi-scale drone targets using a low number of parameters.

**Keywords:** drone object detection; deep learning; lightweight network; coordinate attention



## 1. Introduction

In recent years, drone technology has developed rapidly. However, there are more and more black flying and random flying problems of small drones, which pose a serious threat to both society and individuals [1]. Therefore, it is very urgent to research the defense technology of small drones. Drone targets have the characteristics of low flight altitudes, slow speeds, and miniaturization [2]. This makes radar and radio frequency detection methods very difficult and costly [3,4]. Although the sound detection method is easy, the targets' detection position can be interfered by noise [5]. At present, a new object detection method for drones is very necessary.

With the widespread application of deep learning and the continuous updating of GPUs, object detection has gradually moved from traditional pattern recognition to deep learning and has been widely used in face recognition, medical image detection, automatic vehicle driving, and other engineering tasks [6]. At present, deep learning algorithms can be mainly divided into two-stage object algorithms and one-stage object algorithms. The former extracts target candidate regions through a candidate box generator and then classifies and regresses the candidate boxes. Its representative algorithms include R-CNN and Faster R-CNN [7]. The latter does not need candidate boxes and can directly extract image features through a convolutional network for classification and location. Its representative algorithms include SSD (Single-Shot MultiBox Detector) and YOLO (You Only Look Once) [8]. YOLOv4 is a classic version of the YOLO algorithm series [9]. It is the first to use CSPDarkNet-53 as the backbone to extract feature information, PANet (Path Aggregation Network) as the feature fusion network, and an SPP (Spatial Pyramid Pooling) structure to enhance feature extraction [10], which makes YOLOv4 more effective in multi-scale target detection and greatly improves the performance of one-stage algorithms.

Deep learning algorithms have high accuracy and can implement the detection task of different targets well. Therefore, many scholars use deep learning methods to study drone detection. Fatemeh et al. use a CNN (Convolutional Neural Network) as a classification model to detect drones [11]. The model achieves 93% accuracy in the self-built dataset, which is far better than the SVM (Support Vector Machine) and KNN (K Nearest Neighbor) methods in the comparative experiment. However, the amount of parameters in the convolution model is too large to perform detection tasks quickly, and the detection performance of multi-scale targets is poor. To solve the problem that the convolutional network does not fully extract the feature information of multi-scale targets, Zeng et al. propose a detection algorithm using Res2net combined with a hybrid feature pyramid structure to achieve multi-scale feature fusion, which achieves more than 93% mAP in the self-built drone dataset [12]. However, the complex convolution and multi-scale structure make the model have poor real-time performance. Aiming at the problems of large memory consumption and poor real-time performance of the drone detection model, Tian et al. propose an improved two-scale YOLOv4 [13]. Through model pruning and sparse training, this method reduces the memory occupation by 60%, increases the FPS by 35%, and reaches 58 frames per second. However, the accuracy of the model has decreased. In drone detection, Yew et al. construct a multi-scale drone dataset and integrate SSD and YOLOv3 for training, which greatly improves the confidence of the detection result [14]. However, the integration method brings about the problems of slow speed and a complex process. In the process of improvement, the above methods both have problems balancing the accuracy and speed of their models.

Aiming at the problems of large model sizes and low multi-scale target detection accuracy, we propose a drone target detection algorithm called YOLOv4-MCA that combines the lightweight network MobileViT, coordinate attention, and improved K-means++. The innovations of this paper are as follows:

- The proposed approach uses the lightweight network MobileViT as the backbone of the detection model and adopts depth-wise separable convolution to replace the standard convolution in the feature fusion network and the detection head network. In exchange for a small loss of accuracy, the number of model parameters and the model complexity are reduced, and the detection speeds are improved.
- The proposed approach adopts a new multi-scale attention feature fusion network CA-PANet as the neck of the detection model. It can fuse multi-scale feature information and enhance the flow of high-dimensional texture features and low-dimensional positioning features. The introduced coordinate attention module can extract more positional information. After improvement, the classification and positioning accuracy of multi-scale targets have been improved.
- In the anchor box setting, we introduce the improved cluster algorithm K-means++ to cluster data samples and update the anchor box size to improve detection efficiency.

Through the above improvements, our method has a lighter network structure, faster detection speed, and greater accuracy than other detection algorithms. Thereby, we use YOLOv4-MCA to conduct the drone detection research.

## 2. Related Work

This purpose of this section is mainly to introduce the development and applications of relevant methods, including the development of lightweight networks, the methods of attention mechanisms, and some research work on anchor boxes.

With the development of convolutional networks, models become larger and deeper. Such development brings high accuracy to the model, but it also loses the advantages of smaller sizes and higher speeds. Contrary to large convolutional networks, lightweight networks are efficient models for mobile and embedded vision applications. With simple architectures, small sizes, and short response delays, lightweight networks can better meet the requirements for lightweight and fast models in drone detection tasks. At present, the representative lightweight networks are EfficentNet, MobileNetV3, MobileViT, and so

on [15–17]. With lightweight networks, we can design agile models to maintain a balance between accuracy and speed. However, it is difficult to effectively combine lightweight networks with detection models. In this regard, Zhao et al. conduct relevant research in weed detection and propose an an improved YOLOv4 combined with MobileNetV3 and CBAM [18]. The mAP value of the improved model for weed detection in a potato field reaches 98.52%, and the average detection time of a single image is 12.49 ms, which shows that the improved YOLOv4 model is a feasible real-time weed identification method. In addition to lightweight networks, model pruning is also used to construct drone detection models. In view of small and fast drones, Zhang et al. propose a drone detection method by pruning the convolutional channel and residual structures of YOLOv3-SPP3 [19]. The pruned model achieves good results on the self-built UAV dataset: the maximum detection speed increased by 10.2 times and the maximum mAP value increased by 15.2%, which meets the requirements for the real-time detection of UAVs. Similarly, Liu et al. present a YOLOv4 model for pruning the convolutional channel and shortcut layer to address the threat of small and quick drones [20]. This pruned YOLOv4 model achieves 90.5% mAP, and its processing speed is increased by 60.4%, which is an effective and accurate approach for drone detection. Methods based on a lightweight network and model pruning can obtain remarkable results in the transformation of drone detection models but have the problems of performance degradation and network model instability.

Attention mechanisms have been proven to be helpful for solvingcomputer vision tasks in many research works [21]. Therefore, we can apply attention mechanisms to improve the accuracy of drone detection models. At present, the widely used attention mechanisms include SE (Squeeze and Excitation), CBAM (Convolutional Block Attention Module), CA (Coordinate Attention), and so on [22–24]. In the combination of attention mechanisms and drone-based detection, Li et al. take the lead and propose a novel object detection algorithm for drone cruising in large-scale maritime scenarios [25]. By introducing the self-attention structure Transformer to enhance the feature extraction function, the improved method increases the detection precision by 1.4% and the number of parameters is reduced by 11.6%. In the same way, Li et al. construct an inverted pyramid network based on the spatial attention mechanism to improve the detection performance of small and dense traffic signs [26]. Cao et al. add the CBAM attention module to improve the prediction accuracy of complex small target detection by the YOLOv4 network [27]. These methods of adding attention mechanisms have achieved good results in multi-scale target detection. Thereby, selecting an appropriate attention mechanism is helpful for drone target detection. However, the extraction of location information seems to be neglected, which leads to a poor detection effect for small targets.

In addition updating the network structure, the improvement in the anchor box is also an important part of object detection. From the initial prior knowledge setting, to K-means algorithm clustering, to the present anchor-free approach, anchor box setting in model training and detection is becoming more and more reasonable and efficient. Cai et al. significantly increased the detection speed by setting anchors of different sizes on multi-scale feature maps [28]. Piao et al. propose a two-stage, anchor-free network to predict regression results stage-by-stage, thereby reducing the scope of the prediction space and improving the localization accuracy [29]. Hu et al. propose an adaptive approach based on the ISODATA clustering algorithm to learn the anchor shape priors from data samples, and this method solves the identification problems for small targets owing to the multiple down-samplings performed in a deep-learning-based method [30]. It can be seen from these methods that the improvements in anchor box methods are helpful to improve the detection performance of these models and thus improve the detection performance on drone targets.

Combined with the improvements and deficiencies of the relevant papers, we propose a lightweight drone detection algorithm based on MobileViT and Coordinate Attention, and the improved K-means++ clustering algorithm is used to reset the anchor box.

## 3. Model Design

This section mainly introduces the overall model architecture of YOLOv4-MCA and the principles of the relevant improvement methods.

### 3.1. YOLOv4-MCA Model

YOLOv4-MCA is the core algorithm of this paper. This section will introduce its basic architecture, including its backbone, neck, and prediction head:

(1) Backbone Network: The algorithm uses lightweight MobileViT11 as the backbone feature extraction network. In terms of network structure, the MobileViT backbone prunes the full connection layer and prediction layer and leaves the initial convolution layers, MV2 block and MVIT block. Its main structure is shown in Figure 1. Functionally, the new backbone combines the advantages of CNNs and Vision Transformer. The combined network structure can fully extract the local and global sample information and generate feature maps of different scales to the neck network.

(2) Neck network: The algorithm uses multi-scale attention CA-PANet as the neck feature fusion network. CA-PANet has three feature fusion branches and adds a coordinate attention module to the network input nodes. The multi-scale attention network can promote the extraction of target location information, realize the fusion of deep and shallow feature information, and achieve feature enhancement. Its network structure mainly consists of CBR block, Upsample, Downsample, CA block, and SPP block.

(3) Prediction head: The algorithm uses the classic Yolohead as the prediction head. Its main function is to predict the position and class of the target feature maps. Its main structure is composed of a CBR block and DW convolution block, as well as loss function and prediction box filtering algorithm.

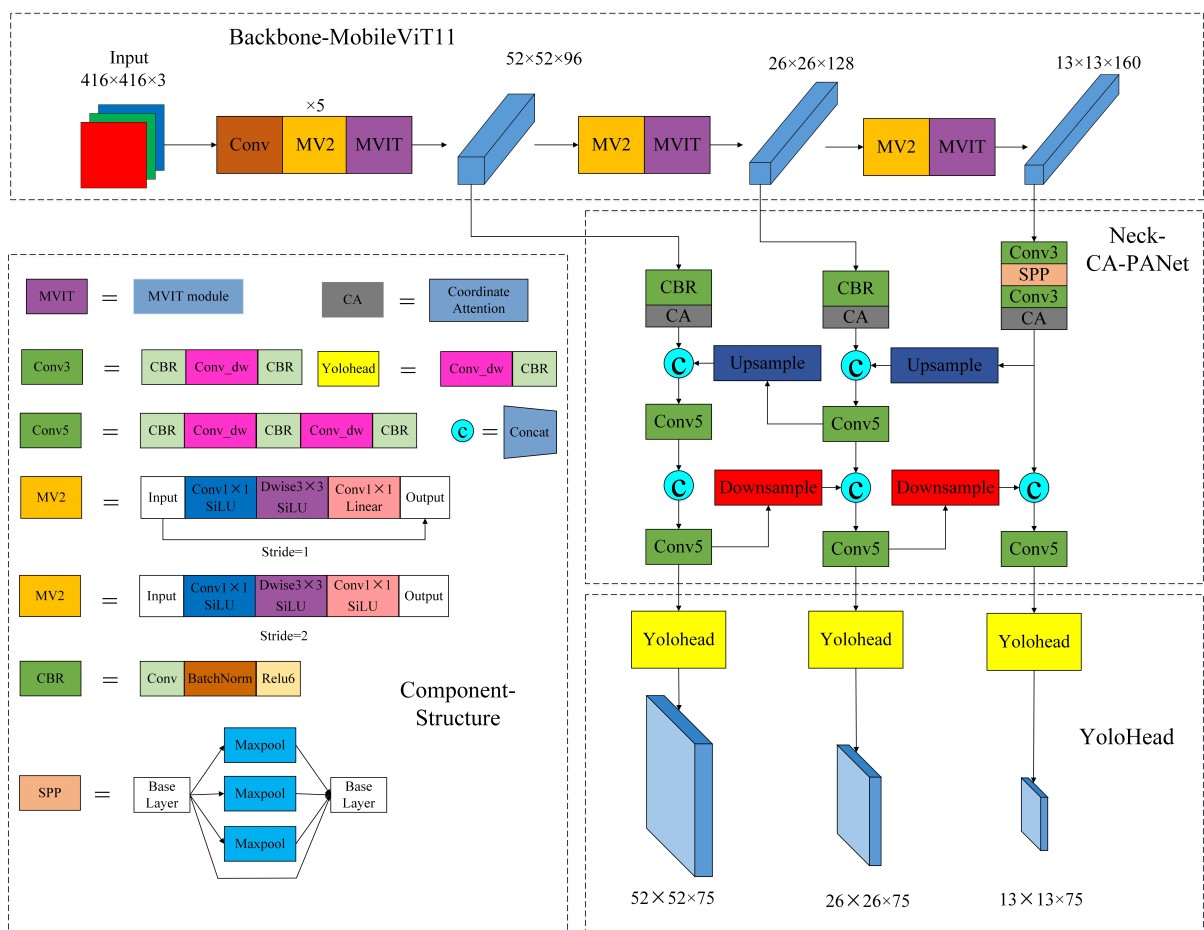

**Figure 1.** YOLOv4-MCA model architecture diagram.

In addition to the three parts, there are other small modules. CBR block is composed of a convolution block, a normalization layer, and an activation function (Relu6). It is the main operator for model extraction of convolution features. The MV2 block is a linear bottleneck inverted residual structure, which is mainly composed of a $1 \times 1$ convolution block, a $3 \times 3$ DW convolution block, and an activation function (SiLU). When the input and output have the same channel, it will make a residual connection to deepen the network. The MVIT block is the main component module of the vision transformer in MobileViT, which is used to extract the global information representation. The CA block is a coordinate attention module. Conv3 and Conv5 are composed of CBR and DW convolution blocks. The SPP block is composed of the maximum pooling layer and the full connection layer. Its function is to convert input feature maps of different sizes into feature vectors of the same size. Figure 1 shows the overall structure of YOLOv4-MCA and the structure of each block.

### 3.2. Lightweight Backbone MobileViT11

To ensure the low parameter demand and real-time detection performance, we use a lightweight MobileViT network to design the backbone MobileViT11. MobileViT is a lightweight, general-purpose, and mobile-friendly vision transformer proposed by Sachin et al. of Apple in 2021 [17]. Different from the single CNN backbone of YOLOv4, MobileViT combines the architectures of CNNs and ViTs. Therefore, it not only has the light weight and efficiency of CNNs, but it also has the self-attention and global vision of transformer networks, which allows it to learn local features and global representations better. Obviously, it is a lightweight model with more comprehensive performance.

As a classification model, MobileViT performs well in extracting feature information. However, when being used as the backbone of the detection model, the network structure needs to be adjusted. Considering that the classification layer and global pooling layer do not participate in feature extraction, we prune these two layers and adjust the output nodes of the model to obtain the MobileViT11 backbone network, which can output the multi-scale feature information maps. Table 1 shows the structure information of the pruning model. In Table 1, Floor represents the layer number of the module in the network, Input represents the size of the input feature map, Operator indicates the type of module, Out size indicates the number of output channels of each layer, L represents the number of transformers in the MVIT block, and s is the step size of convolutional kernel movement.

**Table 1.** The structural information of MobileViT11.

| Floor | Input | Operator | Out Size | L | s |
|-------|-------|----------|----------|---|---|
| 1 | $416^2 \times 3$ | Conv | 16 | - | 2 |
| 2 | $208^2 \times 16$ | MV2 | 32 | - | 1 |
| 3 | $208^2 \times 32$ | MV2 | 64 | - | 2 |
| 4 | $104^2 \times 64$ | MV2 | 64 | - | 1 |
| 5 | $104^2 \times 64$ | MV2 | 64 | - | 1 |
| 6 | $104^2 \times 64$ | MV2 | 96 | - | 2 |
| 7 | $52^2 \times 96$ | MVIT | 96 | 2 | 1 |
| 8 | $52^2 \times 96$ | MV2 | 128 | - | 2 |
| 9 | $26^2 \times 128$ | MVIT | 128 | 4 | 1 |
| 10 | $26^2 \times 128$ | MV2 | 160 | - | 2 |
| 11 | $13^2 \times 160$ | MVIT | 160 | 3 | 1 |

The main components of the MobileViT11 are MV2 block and MVIT block. MV2 is a linear bottleneck inverse residual block proposed in MobileNetV2 [31]. The function of this structure is to expand the low-dimensional compressed data to higher dimensions, filter the data with depth-wise separable convolution, and restore the feature data back to the lower dimensions through the linear bottleneck block. This structure uses small tensor data in the reasoning process, which reduces the demand on the embedded hardware for main memory access and improves the response speed. Its structure is shown in Figure 2.

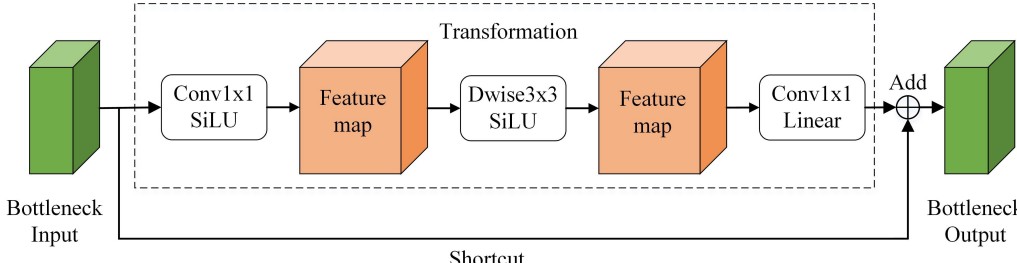

**Figure 2.** The structure of linear bottleneck inverse residual block.

Another block, the MVIT block, consists of three parts: the local information coding module, the global information coding module, and the feature fusion module. The corresponding functions of the three parts are to extract local feature, extract global feature, and fuse feature information, respectively. MVIT can fully extract the image feature information with fewer parameters. The components of the MVIT block are shown in Figure 3.

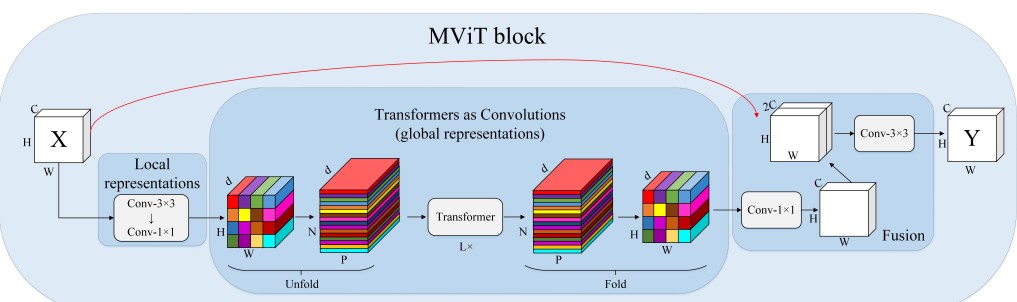

**Figure 3.** The structure of MVIT block.

### 3.3. Multi-Scale Attention CA-PANet

The lightweight model has a simplified architecture, but it also brings about a loss of accuracy. To reduce the decline in model accuracy, we propose a multi-scale attention network CA-PANet, which introduces a coordinate attention module and depth-wise separable convolution into the path aggregation network. The structure of CA-PANet is similar to PANet. It adopts a top-down feature extraction branch and a bottom-up feature enhancement path. Through feature fusion, the network can combine shallow positioning information and deep semantic information to enhance the features richness. At the input node of multi-scale features, coordinate attention blocks are added to group and code the feature map to augment the representation of targets of interest. A lightweight DW convolution block is used to replace the standard convolution to further decrease the model complexity. The activation function uses SiLU instead of ReLU6 to improve the convergence of the deep network model. The structure of the CA-PANet is shown in Figure 1.

The Coordinate Attention module was designed by Hou et al. for their Efficient Mobile Network [24]. It can embed the positional information into channel attention, factorize feature maps into direction-aware and position-sensitive attention maps by encoding. In this way, the network can aggregate features along two spatial directions to generate spatial selective attention maps. The resulting feature maps contain more positional information, which has more advantages in object detection, semantic segmentation, and other tasks. Figure 4 shows the structure of the coordinate attention module.

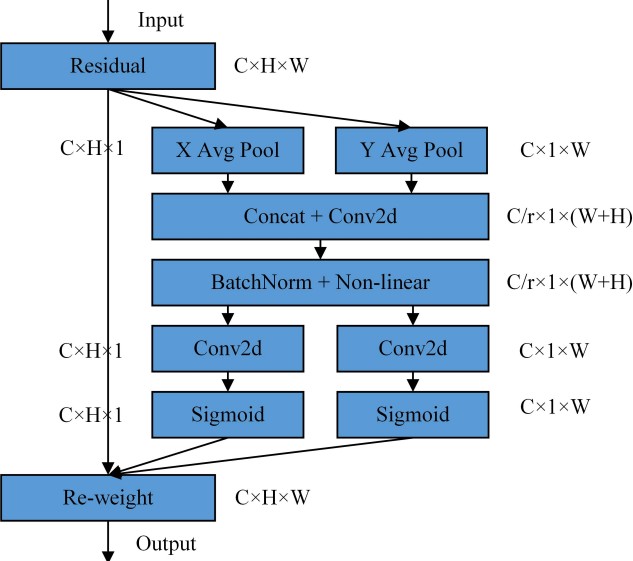

**Figure 4.** The structure of coordinate attention module.

In Figure 4, Coordinate Attention divides feature map calculation into two parts: coordinate information embedding and coordinate information generation. The embedded part can be abstracted into two 1D feature-encoding formulas in the horizontal direction and vertical direction, respectively. Given the input $x$ and the size of $(H, 1)$ or $(1, W)$ pooling kernel, the output of the c-th channel with height $h$ can be formulated as:

$$z_c^h(h) = \frac{1}{W} \sum_{0 \le i < W} x_c(h, i), \tag{1}$$

Similarly, the output of c-th channel with width $w$ can be written as:

$$z_c^w(w) = \frac{1}{H} \sum_{0 \le j < H} x_c(j, w), \tag{2}$$

In the generation part, the input features will be concatenated, which can make full use of the captured positional information and highlight the interesting regions. Finally, after the convolution transformation function, the output y of the coordinate attention block can be formulated as:

$$y_c(i, j) = x_c(i, j) \times g_c^h(i) \times g_c^w(j), \tag{3}$$

where $y_c$ represents the output, $x_c$ represents the input, $c$ represents the c-th channel, and $g_c^h$ and $g_c^w$ represent the attention feature weights in the horizontal and vertical directions respectively.

Depth-wise separable convolution is a convolution block proposed in MobileNetV1 [32]. It is composed of a $3 \times 3$ light depth convolution layer and a $1 \times 1$ heavy point convolution layer, which separates the spatial filtering from the feature generation and greatly reduces the number of convolution kernel operations. The structure of DW convolution is shown in Figure 5.

The SiLU function is an improved version of Sigmoid and ReLU. It has the characteristics of a lower bound but no upper bound and is smooth and non-monotonic. Moreover, SiLU has better optimization performance than ReLU in deep models. Assuming that the input of the activation function is $x$, the formula of SiLU can be written as:

$$SiLU[x] = \frac{x}{1 + e^{-x}}, \tag{4}$$

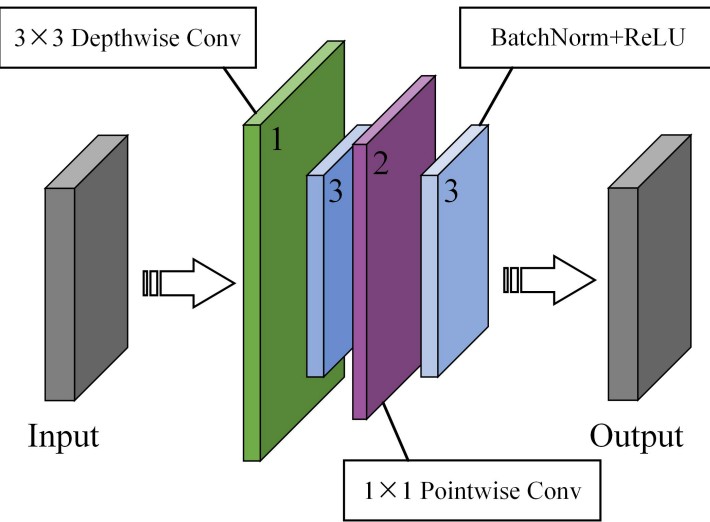

**Figure 5.** The structure of depth-wise separable convolution.

*3.4. Improved K-means++ Anchor Box*

An anchor box is a pixel box used to traverse the region of interest in an image to determine the correct box in object detection. In model training, a reasonably designed anchor box can help the model learn more accurate sample features and improve the accuracy. In the traditional two-stage detection, the anchor box is set according to the prior knowledge, which has a weak adaptive ability and poor learning effect on the diverse scales of distribution samples. In this paper, we use K-means++ to cluster data samples to obtain appropriate anchor boxes. This method has stronger robustness and can be designed for different datasets.

The principle of K-means++ is to select the appropriate distance measure and criterion function for iteration and finally select k clustering center points. Compared with the original K-means algorithm, the improved K-means++ discards the Euclidean distance measure and the method of randomly selecting the initial point, selects the IoU (Intersection over Union) between the background box and the prediction anchor box as the measurement, and finds the points with the longest distance from each other as the cluster center. This method reduces the error in the calculation of different anchor boxes and the possibility of obtaining weak clustering centers. The improved clustering method of K-means++ can be formulated as:

$$f = \arg\max \frac{\sum\limits_{i=1}^{k} \sum\limits_{j=1}^{n_k} I_{IoU}(B, A)}{N},$$ (5)

In Formula (5), $B$ represents the background box, $A$ represents the prediction anchor box, $k$ represents the number of cluster centers, $n_k$ represents the number of background boxes in the k-th cluster center, $N$ represents the total number of background boxes, $I_{IoU(B,A)}$ represents the IoU between the background box and the prediction anchor box, $i$ represents the cluster center number, and $j$ represents the background box number in the cluster center.

Figure 6 shows the visualization results of the improved K-means++ algorithm on Drone-dataset. And the final nine anchor boxes is obtained as (19, 20), (41, 39), (91, 78), (187, 151), (293, 251), (491, 283), (394, 414), (553, 405), and (578, 537).

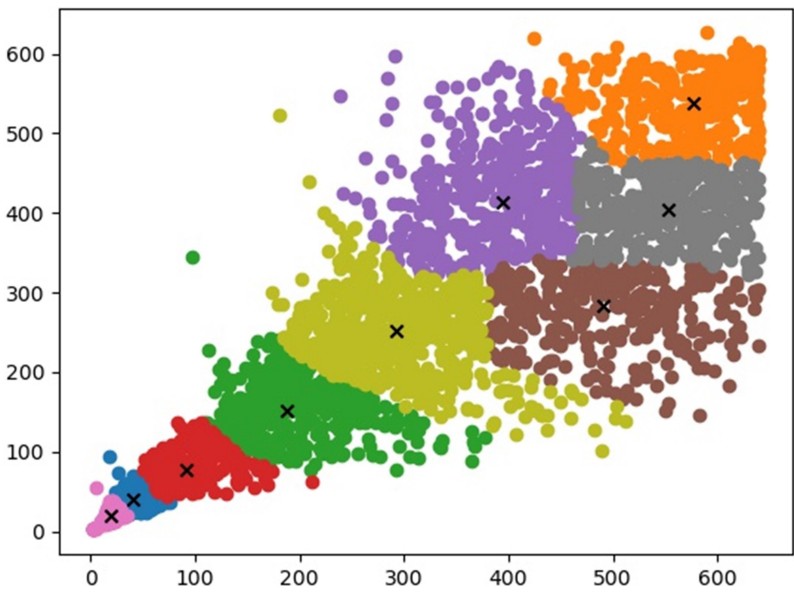

**Figure 6.** The visualization results of improved K-means++.

## 4. Dataset and Experiment Information

### 4.1. Dataset Preparation

The drone dataset used in the experiment is constructed from a public dataset [14] and drone target image expansion. The drone dataset contains large-, medium-, and small-scale drone targets, and there 3813 and 3834 images and targets, respectively. In the dataset division, the ratio of training set and testing set is 4:1, and the ratio of the training set and validation set is 9:1. All samples are annotated by Labelimg software. Table 2 shows the detailed information of Drone-dataset.

**Table 2.** The table of Drone-dataset's information.

| Class | Number | Large | Medium | Small |
|---|---|---|---|---|
| train | 2762 | 1610 | 582 | 570 |
| validation | 306 | 179 | 59 | 68 |
| test | 766 | 442 | 162 | 162 |
| total | 3834 | 2231 | 803 | 800 |

In addition to Drone-dataset, this paper also uses the PASCAL VOC 07+12 dataset to conduct generalization performance experiment. VOC 07+12 is a combined version of PASCAL VOC 2007 and PASCAL VOC 2012, which includes images and labels for object detection, image classification, object segmentation, and action recognition. It has 20 types of targets and 21,504 images, including 16,551 training set images and 4952 test set images for object detection.

To overcome the large difference in target scales between images in the dataset, model training uses the data augmentation method Mosaic. It is used in YOLOv4 to cut and splice pictures. This method can enrich data diversity, increase background complexity, and improve the scale robustness of the model. The following Figure 7 shows the effect of Mosaic.

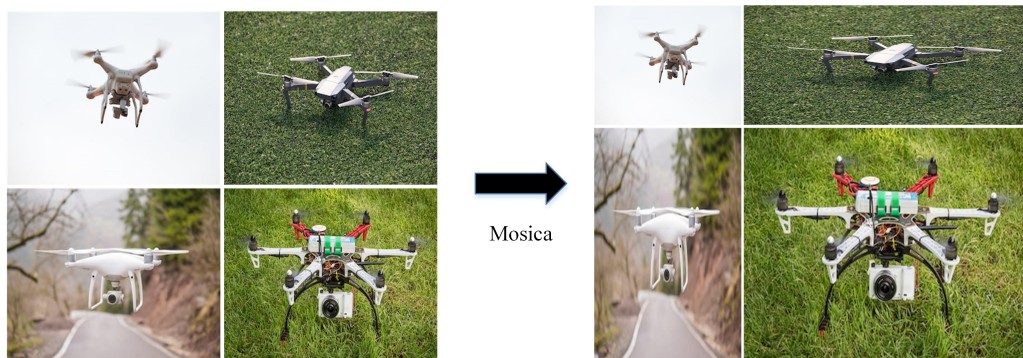

**Figure 7.** Example of Mosaic data augmentation. **Left**: The original image. **Right**: The processed image.

*4.2. Environment and Model Training*

The experimental environment is based on the deep learning framework PyTorch and the parallel computing platform CUDA 10.2. The computer operating system is a Windows 10 operating system with an Intel (R) Core (TM) i9-10900F CPU @ 2.80GHz 2.81 GHz, 64G of memory, and an NVIDIA Quadro P4000 GPU with 8G of video memory.

In order to implement more accurate experiments, we adopt the following model training strategies. First, we pre-train the model in a large sample dataset, PASCAL VOC 07+12. Then, we load the pre-training weights on the model through transfer learning and set the appropriate training hyperparameters. At last, we train and verify the pre-training model on Drone-dataset. Experiments show that this method is much better than training the model from zero. Table 3 shows the hyperparameter information used in model training. Figure 8 shows the trend of the model training loss.

**Table 3.** The hyperparameter setting of model training.

| Type | Parameter | Note |
|------|-----------|------|
| Image size | 416 × 416 | Image input size |
| Epoch | 300 | Total training times |
| Batch size | 16 or 8 | Freeze size or Normal size |
| Learning rate | 0.01 and 0.0001 | Initial and Minimum rate |
| optimizer | SGD | Optimizer type |
| momentum | 0.937 | Momentum of optimizer |
| Weight decay | 0.0005 | The decay of weights |

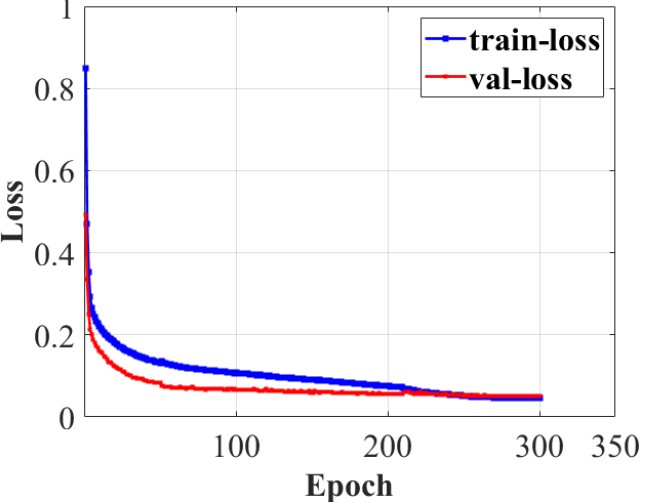

**Figure 8.** The loss value of YOLOv4-MCA on Drone-dataset.

As can be seen from Figure 8, the train loss and validation loss of the model decrease with the increase in epochs. The loss of the first 50 epochs decreases rapidly, indicating that the span of the model weight update is large. The loss from 50 to 200 epochs decreases gently, indicating that the model is in the fine-tuning stage and is constantly approaching the optimal value. In the last 100 epochs, the loss curve is in a stable state, which indicates that the model training reaches saturation, and the model basically achieves its best performance.

*4.3. Evaluation Metric*

The evaluation metrics of the experiment include object detection accuracy AP (average precision), mAP (mean Average Precision), detection speed FPS (Frames Per Second), model parameters, and model volume size.

mAP can comprehensively evaluate the localization and classification effect of the model for multi-class and multi-target tasks. Calculating the mAP requires calculating the AP for each class in the recognition task and then taking its average. The formula is as follows:

$$mAP = \frac{\sum_{i=1}^{C} AP_i}{C}, \tag{6}$$

In Formula (6), *C* represents the number of total classes, and $AP_i$ represents the AP value of class *i*.

Calculating AP requires the values of P (Precision) and R (Recall). The formulas for these three metrics are as follows:

$$P = \frac{TP}{TP + FP}, \tag{7}$$

$$R = \frac{TP}{TP + FN}, \tag{8}$$

$$AP = \int_0^1 P(R)dR, \tag{9}$$

In Formulas (7)–(9), $TP$ (True Positive) means that the input is a positive sample and the predicted result is also a positive sample; $FP$ (False Positive) means that the input is a negative sample and the predicted result is a positive sample; $FN$ (False Negative) means that the input is a positive sample and the prediction result is a negative sample; $TN$ (True Negative) means that the input is a negative sample and the prediction result is a negative sample.

The FPS metric is the time that a model takes to detect a picture or the number of pictures detected in one second. The larger the FPS, the faster the model is detecting targets, which can be used to measure the detection speed of model. The model parameters and model volume size are both metrics of model complexity. They all represent the size of the model, which can directly reflect model size.

## 5. Experimental Results and Analysis

In this section, we conduct a series of experiments to demonstrate the performance of the proposed approach. Firstly, we perform a general method experiment based on PASCAL VOC 07+12 to investigate the advantages of our method compared with other common algorithms. Next, we utilize the proposed approach on Drone-dataset as a validity experiment to observe the effectiveness of drone target detection. Finally, we conduct the ablation experiment of YOLOv4-MCA on Drone-dataset. The experiment takes YOLOv4 as the baseline and combines different improvement strategies to explore the contribution of each improved components in the proposed approach.

### 5.1. The Universality Experiment

To demonstrate the general performance of YOLOv4-MCA, we conduct an experiment using Faster R-CNN, SSD, YOLOv3, YOLOv4, YOLOv5-m, and YOLOv4-MCA. All algorithms are trained and validated on the PASCAL VOC07+12 dataset. The metrics used in the experiment are mAP (IoU = 0.5), FPS, model parameters, and model volume size. In this way, we can compare the advantages and disadvantages of the model in all aspects. Table 4 shows the detection results of each algorithm on PASCAL VOC 07+12.

**Table 4.** The performance comparison of algorithms on PASCAL VOC 07+12.

| Model | Backbone | Input Size | mAP (%) | FPS (f/s) | Parameter ($\times 10^6$) | Volume (MB) |
|---|---|---|---|---|---|---|
| Faster R-CNN | ResNet50 | $416 \times 416$ | 77.02 | 12 | 137.10 | 522.99 |
| YOLOv4 | CSPDarkNet53 | $416 \times 416$ | 84.29 | 26 | 63.94 | 243.90 |
| YOLOv3 | DarkNet53 | $416 \times 416$ | 80.24 | 35 | 61.63 | 235.08 |
| SSD | VggNet16 | $300 \times 300$ | 78.27 | 43 | 26.29 | 100.27 |
| YOLOv5-m | Focus-CSPNet | $640 \times 640$ | 80.46 | 35 | 20.95 | 79.91 |
| YOLOv4-MCA | MobileViT11 | $416 \times 416$ | **80.70** | **40** | **13.47** | **51.39** |

From the experimental results in Table 4, the mAP of our approach is 80.70%, which has the highest accuracy of all algorithms except YOLOv4. This indicates that our method has not significantly reduced the performance of model detection due to its light weight after various improvements. In FPS, our approach improves the model by 14 f/s compared to YOLOv4, which is the same as SSD and better than other algorithms. The lightweight method give the model a faster detection speed, which has more advantages in real-time detection tasks. In model complexity, the proposed algorithm achieves the best performance in model parameters and model volume. It has one-fifth of the model parameters of YOLOv4, which shows that our model is very friendly to the migration of mobile devices.

To sum up, the proposed approach combines the lightweight MobileViT network and the improved multi-scale attention network CA-PANet. These improvements greatly reduce the number of parameters, accelerate the detection speed, and achieve good performance metrics on the PASCAL VOC 07+12 dataset. Compared with one-stage or two-stage detection algorithms such as YOLOv3, YOLOv4, and Faster R-CNN, YOLOv4-MCA shows better performance, fully demonstrating the universality of the algorithm.

### 5.2. The Validity Experiment

To investigate the performance of our approach in drone detection task. We perform the model validity experiment according to the universality experiment. Similarly, all algorithms used in the experiment are trained and verified on Drone-dataset. In addition, we adopt more metrics such as $AP_{Small}$, $AP_{Medium}$, and $AP_{Large}$ (IoU = 0.50:0.95), which can help us to observe the effectiveness of the algorithm on multi-scale targets and better evaluate the model's performance. Table 5 shows the detection results of each algorithm on Drone-dataset. Figure 9 shows the drone detection comparison results of YOLOv4 and the proposed algorithm.

**Table 5.** The performance comparison of algorithms on Drone-dataset.

| Model | mAP | $AP_{Small}$ | $AP_{Medium}$ | $AP_{Large}$ |
|---|---|---|---|---|
| YOLOv4 | 92.45 | 25.10 | 39.42 | 57.05 |
| YOLOv5-m | 91.82 | 30.80 | 41.58 | 65.07 |
| YOLOv3 | 90.88 | 25.10 | 38.00 | 57.93 |
| Faster R-CNN | 88.26 | 12.37 | 39.78 | 62.45 |
| SSD | 86.62 | 8.77 | 38.24 | 65.24 |
| YOLOv4-MCA | 92.81 | 24.65 | 38.26 | 58.46 |

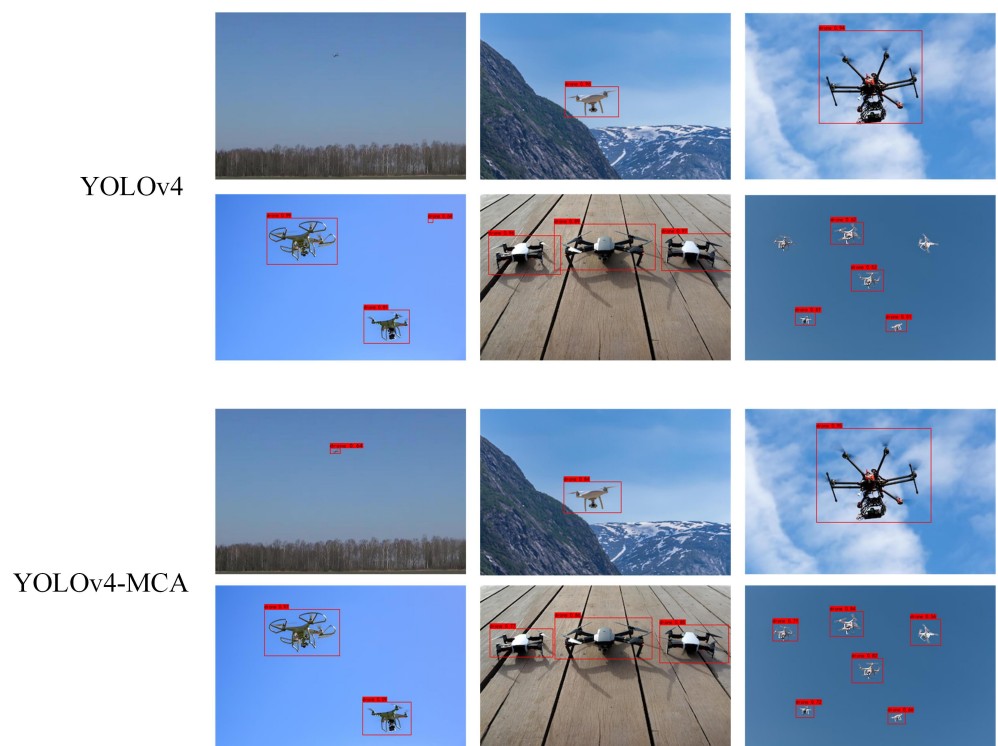

**Figure 9.** The comparison results of YOLOv4 and YOLOv4-MCA.

As can be seen from Table 5, the mAP accuracy of our approach at IOU = 0.5 reaches 92.81%, which is the best result among all algorithms and is 0.36% higher than YOLOv4. The proposed model also performs well on the AP accuracy metrics of three scales. The AP accuracy on small-scale, medium-scale, and large-scale targets is 24.65%, 38.26%, and 58.46%, respectively. In particular, the $AP_{Small}$ of small-scale target detection is 15.88% and 12.28% higher than SSD and Faster R-CNN, respectively. This indicates that the coordinate attention mechanism has a good effect on the extraction of target positional information and enhances the detection ability of small targets. In medium- and large-scale target detection, the proposed model has also maintained good performance and has higher $AP_{Medium}$ and $AP_{Large}$ accuracy. After comparing the results, we argue that the MobileViT network combines the advantages of the ViTs model to enhance the extraction of global features, thus improving the detection performance of the model. In addition, it can be seen in Figure 9 that our approach detects more targets than YOLOv4 and effectively avoids the problems of false detection and missing detection, achieving an overall better detection effect.

In the drone target detection task, the detection object contains more small-scale targets, and has higher requirements on the real-time performance of the algorithm. Therefore, it is necessary to comprehensively consider all aspects of the model's metrics for comparison. Considering the detection accuracy, detection speed, and model complexity, our approach has better performance than other algorithms, which reflects the validity of YOLOv4-MCA for drone target detection tasks.

### 5.3. Method Ablation Experiment

In order to explore the effectiveness of each improved component in the proposed approach, we conduct an ablation experiment. In the experiment, we train each part of the improved component of YOLOv4-MCA on Drone-dataset and record its mAP (IoU = 0.5), FPS, and model parameters metrics to observe their improvement effects. Table 6 shows the metric comparison of the improvement component on Drone-dataset.

**Table 6.** The ablation experiment based on Drone-dataset.

| Model | CSPNet53 | MobileViT | PANet | CA-PANet | Kmeans++ | mAP | FPS | Parameter |
|-------|----------|-----------|-------|----------|----------|-----|-----|-----------|
| 1 | ✓ | | ✓ | | | 92.45 | 26 | 63.94 |
| 2 | | ✓ | ✓ | | | 89.83 | 41 | 13.43 |
| 3 | ✓ | | | ✓ | | 93.16 | 25 | 64.08 |
| 4 | ✓ | | ✓ | | ✓ | 93.85 | 27 | 63.94 |
| 5 | | ✓ | | ✓ | ✓ | 92.81 | 40 | 13.47 |

The models 1 to 5 in Table 6 represent YOLOv4, YOLOv4+MobileViT, YOLOv4+CA-PANet, YOLOv4+Kmeans++, and YOLOv4-MCA, respectively. YOLOv4+MobileViT is the model with lightweight trunk and neck, and YOLOv4+CA-PANet means that coordinate attention is added to PANet. CSPNet53 is the abbreviation of CSPParkNet53.

According to the results in Table 6, the mAP accuracy, FPS, and model parameters of YOLOv4 algorithm are 92.45%, 26 f/s, and 63.94 M, respectively. Compared with YOLOv4, our approach performs better in these metrics. Among them, the lightweight improvement based on the backbone has a relatively obvious improvement effect on the model. At the cost of reducing the mAP by 2.62%, the FPS is improved to 41 f/s and the number of model parameters is reduced to 13.43M. The improvement based on coordinate attention mechanism also performs well, improving the mAP accuracy of the model by 0.71%, but the FPS decreases by 1 frame/s and the number of parameters increase by 0.03M. The anchor box clustering method based on improved K-means++ improves the mAP of the model by 1.4% and the detection speed of 1 f/s. Finally, compared with YOLOv4, the mAP of YOLOv4-MCA reaches the highest value at 92.81%, an increase of 0.36%. The FPS reaches 40 f/s, increasing by 14 f/s. The number of parameters reduces to 13.47M, which is only one-fifth of the number of parameters in YOLOv4. This shows the efficient balance of YOLOv4-MCA in terms of accuracy, speed, and model complexity and also reflects its excellent performance in drone target detection tasks.

## 6. Conclusions

Aiming at the problem of a large number of parameters in detection models and the difficulty of detecting multi-scale drone, we adopt a variety of improved methods on the basis of YOLOv4 and present a novel drone target detection algorithm named YOLOv4-MCA. First, we use the pruned MobileViT lightweight network as the backbone feature extraction network to simplify the model complexity and improve the detection speed. Secondly, we utilize the improved multi-scale attention CA-PANet as the feature fusion network to enhance the extraction of location information and promote the fusion of information from low- and high-dimensional features, thereby enhancing the identification and robustness of multi-scale targets. Finally, we adopt the improved K-means++ clustering method to cluster the target dataset, optimize the anchor box parameters, and improve the detection efficiency. The experimental results demonstrate that our approach performs well on our Drone-dataset and the PASCAL VOC 07+12 dataset at various target scales. Our proposed YOLOv4-MCA provides a practical and feasible research idea for the fast detection of drone targets.

**Author Contributions:** Conceptualization, Q.C., X.L., and B.Z.; methodology, Q.C.; software, Q.C.; validation, Q.C.; formal analysis, Q.C. and X.L.; investigation, Q.C.; resources, X.L. and B.X.; data curation, Q.C.; writing—original draft preparation, Q.C.; writing—review and editing, Q.C., X.L. and Y.S.; visualization, Q.C.; supervision, X.L.; project administration, B.Z. and Y.S.; funding acquisition, B.Z. and Y.S. All authors have read and agreed to the published version of the manuscript.

**Funding:** This research was funded by the Theoretical Research Project (KY20S011) and the Fusion Special Project of Anhui Province (KY21C008), China.

**Data Availability Statement:** The data used to support the findings of this study are available from the corresponding author upon request.

**Acknowledgments:** The author would like to thank all contributors to this study.

**Conflicts of Interest:** The authors declare no conflict of interest.

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
