# Peer review of "Drone Detection Method Based on MobileViT and CA-PANet"

_electronics, doi:10.3390/electronics12010223_

Round 1

Reviewer 1 Report

This manuscript proposes a drone detection model using MobileViT and CA-PANet. Even though the manuscript is significantly described, the model and visual representation are impressive. But in my opinion, some improvements are also needed for better understanding for readers are as follows

l   At the end of the introduction, the author must mention the reason why they choose the YOLOv4-MCA over the others like YOLOv5 and YOLOv7.

l   Figure 1 shows the backbone structure of MobileViT11. And the size of the backbone output

feature map is 52×52×40, 26×26×112, 13×13×160. But this size information is not consistent with the introduction in Table 1. Please explain this problem.

l   In Figure 1, the activation function used in the MV2 block is SiLU. However, in the linear bottleneck inverse residual block structure, please explain this problem.

l   The font format and graphic size in Figure 8 are not appropriate. It is suggested to adjust it.

l   It is suggested to capitalize the first letter of words in the titles of Sec 4.3, 5.1, 5.2, and 5.3.

Reviewer 2 Report

Dear Authors,

I think this is an exciting and meaningful research. To achieve a lightweight model and fast detection of drones,  an improved MobileViT based YOLOv4 combined with CA-PANet has been proposed in this paper. I would recommend it if the below issues are handled well.

1) In the part of related work, explain more about how these works could be helpful to drone detection which should be the topic of this part. This is not a review paper of existing object detection methods or mechanisms.

2) In Table 4, please use the bold font for the best result.

3) I did not get the meaning of figure 9. What do you want to illustrate by these figures?

Author Response

请看附件。
